# Increasing Trend in Violence-Related Trauma and Suicide Attempts among Pediatric Trauma Patients: A 6-Year Analysis of Trauma Mechanisms and the Effects of the COVID-19 Pandemic

**DOI:** 10.3390/jpm13010128

**Published:** 2023-01-09

**Authors:** Cecilia Maina, Stefano Piero Bernardo Cioffi, Michele Altomare, Andrea Spota, Francesco Virdis, Roberto Bini, Roberta Ragozzino, Federica Renzi, Elisa Reitano, Lucia Corasaniti, Francesco Macchini, Osvaldo Chiara, Stefania Cimbanassi

**Affiliations:** 1Acute Care Surgery and Trauma, ASST-GOM Niguarda, Piazza Ospedale Maggiore 3, 20162 Milan, Italy; 2General Surgery, I.R.C.C.S. San Raffaele Hospital, Via Olgettina 60, 20132 Milan, Italy; 3Department of Surgical Sciences, Sapienza University of Rome, Piazzale Aldo Moro 5, 00185 Rome, Italy; 4Division of General Surgery, Department of Translational Medicine, Maggiore Della Carità Hospital, University of Eastern Piedmont, Corso Giuseppe Mazzini 18, 28100 Novara, Italy; 5Department of Pediatric Surgery, ASST-GOM Niguarda, Piazza Ospedale Maggiore 3, 20162 Milan, Italy; 6Department of Pathophysiology and Transplantation-State University of Milan-Acute Care Surgery and Trauma, ASST-GOM Niguarda, Piazza Ospedale Maggiore 3, 20162 Milan, Italy

**Keywords:** pediatric trauma, trauma intentionality, trauma mechanisms, COVID-19, suicide attempt, violence-related trauma

## Abstract

Background: Trauma is the leading cause of morbidity and mortality in the pediatric population. During the COVID-19 pandemic (COVID-19), different trends for pediatric trauma (PT) were described. This study aims to explore the trend over time of PT in our center, also considering the effects of COVID-19, focusing on trauma mechanisms, violence-related trauma (VRT) and intentionality, especially suicide attempts (SAs). Methods: All PT patients accepted at Niguarda Trauma Center (NTC) in Milan from January 2015 to December 2020 were retrospectively analyzed. We considered demographics and clinical variables and performed descriptive and year comparison analyses. Results: There were 684 cases of PT accepted at NTC: 84 in 2015, 98 in 2016, 125 in 2017, 119 in 2018, 114 in 2019 and 144 in 2020 (*p* < 0.001), 66.2% male, mean age 9.88 (±5.17). We observed a higher number of traffic-related, fall-related injuries and an increasing trend for VRT and SAs, peaking in 2020. We report an increasing trend over time for head trauma (*p* = 0.002). The Injury Severity Score did not significantly change. During COVID-19 we recorded a higher number of self-presenting patients with low priority codes. Conclusions: NTC is the adult level I referral trauma center for the Milan urban area with pediatric commitment. During COVID-19, every traumatic emergency was centralized to NCT. In 2020, we observed an increasing trend in SAs and VRT among PT patients. The psychological impact of the COVID-19 restriction could explain this evidence. The long-term effects of COVID-19 on the mental health of the pediatric population should not be underestimated. Focused interventions on psychological support and prevention of SAs and VRT should be implemented, especially during socio-demographic storms such as the last pandemic.

## 1. Introduction

Trauma is the leading cause of morbidity and mortality in the pediatric population [1,2]. Over the last few years, changes in trauma intentionality and mechanisms have been reported, noticing an increasing trend in violence-related trauma (VRT), considering both inflicted violence and self-harm [3,4]. During the COVID-19 pandemic (COVID-19), some modifications in pediatric trauma (PT) patterns were described, mainly ascribed to stay-at-home regulations [2,5,6,7,8]. Social isolation, financial problems and loss of support networks were considered risk factors for pediatric traumas. In Italy, these confinement rules were particularly severe and prolonged [7]. The aim of this study was to analyze the trend over time of PT in a level I trauma center with pediatric commitment in Milan, Northern Italy, focusing on changes in trauma mechanisms, intentionality and COVID-19 effects.

## 2. Materials and Methods

A retrospective, observational cohort study was conducted through a complete data review of consecutive pediatric trauma patients (under 18 years old) referred to Niguarda Trauma Center in Milan between January 2015 and December 2020. Demographic; age; sex; mechanism and intentionality of trauma; type of emergency department (ED) access and triage code; clinical, pre-hospital, and in-hospital vital signs; shock index (SI) [9,10]; therapeutic variables; damage control strategy (DCS); interventional radiology; and early total care treatment were compared among single years to explore trends over time and between pre-(2015–2019) and post-COVID-19 outbreak (2020). Traumatic mechanisms were classified as follows: car accident, motorbike accident, bicycle accident, electric scooter accident, pedestrian, fall-related injuries, stab wound, assault, sports traumas and miscellaneous. Trauma intentionality was categorized into three classes: unintentional, violence-related and self-inflicted. Upon ED arrival, a 4-level triage coding system (white–green–yellow–red) was applied in our center [11]; the white code was never used for trauma patients, since they are always at least a green code for mechanism or suspected injuries. Trauma severity was expressed using the Injury Severity Score (ISS), with major trauma defined by an ISS ≥ 12, based upon the organ-specific Abbreviated Injury Scale 2015 Revision (AIS15) [12]. This scale was used to describe the anatomical distribution of injuries in major trauma; an AIS15 score > 2 recognized a severe injury [13]. All patients were managed according to international guidelines [10,14,15,16] and internal protocols [17]. In unstable patients, a DCS approach [18] was used. Emergency surgery was performed on stable patients with trauma-related acute surgical conditions.

### Statistical Analysis

Data were recorded in a computerized spreadsheet (Microsoft Excel 2016; Microsoft Corporation, Redmond, WA, USA) and analyzed with statistical software (IBM Corp., released 2012, IBM SPSS Statistics for Windows, Version 21.0; Armonk, NY, USA). Data are expressed as n (%), mean values ± standard deviation (SD), and median (interquartile range (IQR)). The sample distribution was evaluated with the Kolmogorov–Smirnov and Shapiro–Wilk tests. Continuous variables were compared using a one-way ANOVA test, while categorical variables were analyzed using Pearson’s chi-squared test. The comparison between years was performed with the Mantel–Haenszel test to explore the linear-by-linear association. Moreover, we performed a subgroup comparison considering patients younger and older than 13 years to explore possible different patterns of trauma mechanisms and injuries between the two groups.

A *p*-value less than 0.05 was considered significant.

## 3. Results

### 3.1. Epidemiological Analysis over Time

During the study period, six hundred eighty-four cases of PT were accepted at the Niguarda Trauma Center, with a rising trend over time (Figure 1).

Description: an increasing overall trend of PT was reported in the study period, with a fluctuation of the percentage of major trauma peaking in 2019.

Sixty-six percent were male, with a mean age of 9.88 (±5.17) years. Table 1 shows the baseline population characteristics over time.

No differences in gender distribution, mean age and type of trauma—blunt vs. penetrating—arose across the study period. A significant growing percentage of VRT and suicide attempts (SAs) was observed during the study period, from 3.6% (n = 3) to 9.8% (n = 14) and from 1.2% (n = 1) to 6.2% (n = 9), respectively (*p* = 0.020 and *p* = 0.029) (Figure 2).

We found an increasing trend of VRT and SAs in the overall population over time. Moreover, we show how the most VRT and SAs happened in the teenager subgroup (13–18 years).

Among SA patients (n = 28), fall-related injuries were the predominant mechanisms (92.9%, *p* < 0.001) (Figure 3).

From the comparison analysis between the pre-pandemic group (2015–2019) and the COVID-19 group (2020), no differences in trauma intentionality or mechanisms were found (*p* = 0.780 and *p* = 0.339), even though the highest rate of VRT and SAs was observed in 2020. The main change recorded during the pandemic period concerned the type and priority code of ED access, with a higher number of self-presenting patients accounting for a higher number of green codes at triage in detriment of red ones.

Median patients’ vital signs at first pre-hospital detection and on arrival at the ED are depicted in Table 2.

### 3.2. Major Trauma: Trend over Time and Predominant Mechanisms

The mean ISS was 7.6 (±11), with less than twenty percent of major trauma (n = 130, 19%), fluctuating over the study period and peaking in 2019 (n = 29, 22.3% *p* = 0.031) (Figure 1). Major traumas were observed mostly after fall and motorbike accidents (*p* = 0.022 and *p* = 0.046, respectively). The distribution of injuries with an AIS15 > 2 in major trauma was: traumatic brain injuries (TBI) on top (n = 64, 49.2%), followed by thoracic trauma (n = 62, 47.7%), extremities (n = 42, 32.3%) and abdominal injuries (n = 29, 22.3%). Table 3 shows the anatomical distribution of major lesions per trauma mechanism.

We observed an increasing percentage of TBIs with AIS15 > 2 over the study period (from 23.1% in 2015 to 33.3% in 2020, reaching 44.1% in 2019; *p* = 0.002). Three deaths (0.4%) occurred in the study period, two early deaths (<3 h) due to irreversibly compromised hemodynamic status, and one delayed (>24 h) brain death caused by major traumatic brain injury.

### 3.3. Emergency Department Management

In total, 16 patients (2.3%) needed a DCS procedure [18]: 1 (6.3%) emergent thoracotomy for a penetrating injury with foreign body retention; 1 (6.3%) emergent thoracostomy for tension pneumothorax; 2 (12.5%) cases of extraperitoneal pelvic packing (EPP) for uncontrollable bleeding caused by unstable pelvic ring fracture; and 12 (75%) emergent laparotomies, 1 (8.3%) for an abdominal impalement injury, 4 (33.3%) for the hemoperitoneum (1 hepatic laceration, 3 splenic laceration, 1 major vessel laceration), 3 (25%) for intestinal perforation caused by penetrating injury, 1 (8.3%) for diaphragmatic rupture and 3 (25%) for intestinal perforation associated with blunt abdominal trauma. Sixteen (2.3%) patients required a thoracic drain positioning for pneumothorax and/or hemothorax.

### 3.4. Surgical Intervention

In total, 120 patients (17.5%) underwent emergency surgery: 18 neurosurgical procedures; 104 orthopedic surgeries; 9 abdominal surgeries; and 21 other specialistic surgeries, including maxilla-facial, vascular and plastic surgery. Surgically treated patients had a mean age of 12.4 (±4.2) and a mean ISS of 18.4 (±15.7), significantly higher than the overall population (*p* < 0.001 in both cases). The need for urgent surgical intervention appeared to be the highest in self-inflicted traumas (*p* = 0.001). However, observing trauma mechanisms, the only ones related to a higher request for emergency surgery were traffic-related injuries (car accidents *p* = 0.039, motorbike accidents *p* = 0.001).

### 3.5. Subgroup Analysis: 0–12 Years Old vs. 13–18 Years Old

A subgroup comparison analysis was performed to investigate trauma characteristics between children (n = 417, 60.9%) and teenagers (n = 267, 39.1%). Fall-related injuries; traffic-related injuries, both car and motorbike accidents; and stab wounds were more frequent among teenagers (*p* < 0.001, *p* = 0.008, *p* < 0.001 and *p* = 0.003 respectively). Considering VRT, we observed a higher rate of SAs with fall-related injuries among teenagers (*p* < 0.001). During the study period, we found a significantly increasing trend in VRT and SAs among the same age group, with *p* = 0.012 and *p* = 0.018, respectively, (Figure 2). The complete analysis results are reported in Table 4.

Teenagers had higher ISS than children (10.9 ± 13.6 vs. 5.5 ± 8.3, *p* < 0.001), corresponding to higher organ-specific AIS15 of head and extremities (Table 5). Emergency surgery was performed in sixty-nine (25.7%) over-thirteen years old patients against fifty-one (12.3%) children of the younger group (*p* < 0.001).

## 4. Discussion

Niguarda Trauma Center is a referral level I trauma center with pediatric commitment. During the study period, an increasing trend in VRT and SAs was recorded, with a boost related to COVID-19 spreading. Over the last decades, a similar trend was reported in the literature for adult and childhood VRT [19,20,21,22], highlighting a general and progressive psycho-social imbalance; the COVID-19 pandemic represented the final blow on such an unstable background. Sherman et al. [19] observed a growing proportion of penetrating trauma during the lockdown period in New Orleans’ level I trauma center and a more significant proportion of non-accidental injuries compared to previous years. Huge variability in the trauma mechanisms of SAs are reported in the literature, influenced by cultural background: in North America, where firearms are of easy access, the first mechanism of self-inflicted injuries is a gunshot, followed by hanging [23,24]; in our study, the primary mechanism of a SA was a fall from height. In every country, the home was the primary setting of self-violence, adding reliability to the hypothesis of a COVID-19 pandemic worsening psycho-social discomfort. During the COVID-19 pandemic, all EDs of Northern Italy registered a reduced incidence of PT due to the stay-at-home regulations [2,5,8]. At the same time, our institute faced a higher number of pediatric accesses. When the COVID-19 pandemic reached its peak, pressure on the National Health System forced it to modify the allocation of resources [6,7,25]: the majority of hospitals in the urban area of Milan were dedicated to COVID-related emergencies, while all traumatic emergencies were referred to our Center. Such changes could justify the modifications, bucking the global tendency of less but more severe PT [2,5,6], in type and priority codes of ED accesses registered in the pandemic period in our center, together with the reduction in mean ISS after its peak in 2019.

Approximately one-fifth of traumas presented an ISS ≥ 12 in our series, with fall- and traffic-related injuries as the most frequent mechanisms [3,4,26,27]. According to the literature [26,27,28], TBI appeared to be the leading injury in major trauma. We also noticed an increasing number of major head traumas over the study period, justified by the simultaneous rise in fall-related injuries.

Almost twenty percent of our patients underwent invasive procedures, considering both DCS, 2.3% of the population, and urgent surgical interventions; few studies have reported surgical and DCS data on PT [3,28,29]. Ringen et al. [28] recently reported a 0.1% rate of DCS and a 0.8% mortality rate over nine years (2010–2018) in a Norwegian level I trauma center, with no ED thoracotomies in non-survivors after 2013. Despite a slightly lower mortality rate (0.4%), we recorded a higher percentage of DCS, with no DCS applied in early deaths. In general, DCS and surgical outcomes fall outside our study’s aim; future research on the theme is awaited.

From the comparison between younger and older than 13 years old PT patients, a higher proportion of VRT and SAs came up in the older group, with an increasing number over the study period, leading to VRT and SAs occurring exclusively in teenagers in 2020 at NTC. The trend in SAs among adolescents has been growing for decades [30], representing one of the leading causes of death [23,31]. However, we observed a spike in self-inflicted trauma in the last year of the study, probably pushed by quarantine discomfort. We also reported a first small spike in VRT and SAs during the first year due to the identification of our institution as a level I trauma center with pediatric commitment due to the fast-growing experience of our trauma team.

The societal changes dictated by COVID-19 impacted pediatric mental health, child neglect and the occurrence of PT [32]. Epidemiological analyses are valuable for scientific and socio-political communities to identify critical issues and address interventions and control measures. Sweden and Germany took advantage of data from clinical and public health studies, reducing pediatric injuries-related deaths by over 50% by implementing cost-effective prevention strategies such as expanded pre-school services and mandatory swim training [4,33]. The present study evaluates the epidemiological trend in PT, reporting an increasing number of SAs among teenagers related to life-threatening trauma requiring urgent invasive interventions and highlighting a substantial psychological impact of COVID-19 regulations. Its most significant limitation is represented by its retrospective design based upon a single adult level I trauma center. For this reason, our data could not portray the general pediatric population. The centralization of PT during the COVID-19 health system emergency brought power and weakness to our research: on the one hand, it brought to light the psycho-social threat of increasing SAs among teenagers, and on the other carried a vast number of confounding minor trauma patients, usually afferent to peripheral hospitals.

## 5. Conclusions

An increasing trend of VRT and SAs was recorded in our series in accordance with the international literature [23,30,31], representing the mechanisms associated with the most severe trauma and a higher request for interventions [3,4]. We noticed a speed-up in the rise of VRT and SAs during the COVID-19 outbreak and its restrictive regulations. These alarming results express psycho-social disorders, which have been underestimated during the last decade, hidden below the ash that the social isolation of the COVID-19 pandemic has blown on fire. While exacerbating such negative consequences for mental health, the pandemic also offers us an opportunity to rethink our approach and to build back better by investing in a comprehensive approach to mental health that is fit for the future, as proposed by UNICEF World’s Children Report 2021 [34] and the EAP and ECPCP joint statement [35]. Epidemiological research should trigger political and social interventions in prevention and education to reverse this trend.

## Figures and Tables

**Figure 1 jpm-13-00128-f001:**
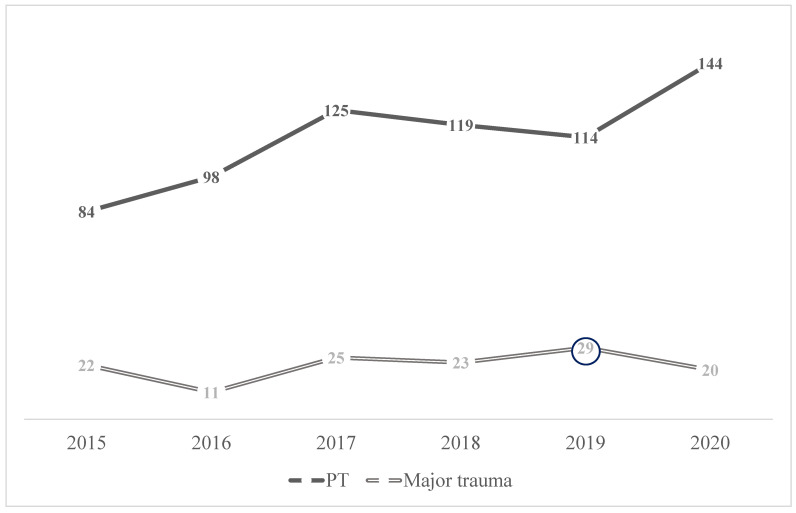
Distribution of overall PT and major trauma (ISS ≥ 12) over time.

**Figure 2 jpm-13-00128-f002:**
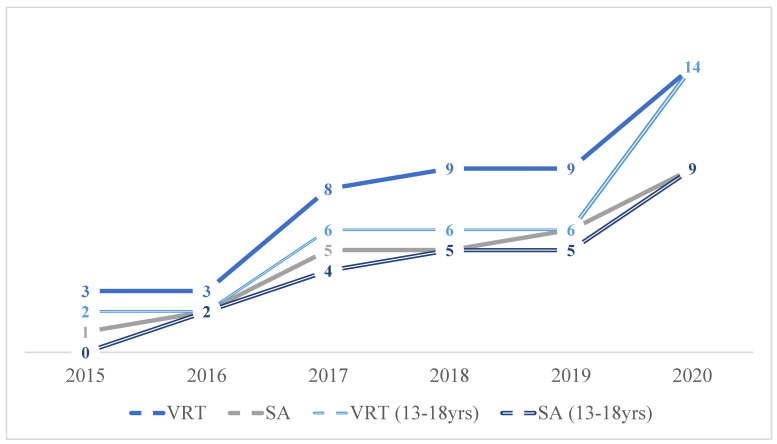
The trend of VRT and SAs in the overall population and the teenager subgroup.

**Figure 3 jpm-13-00128-f003:**
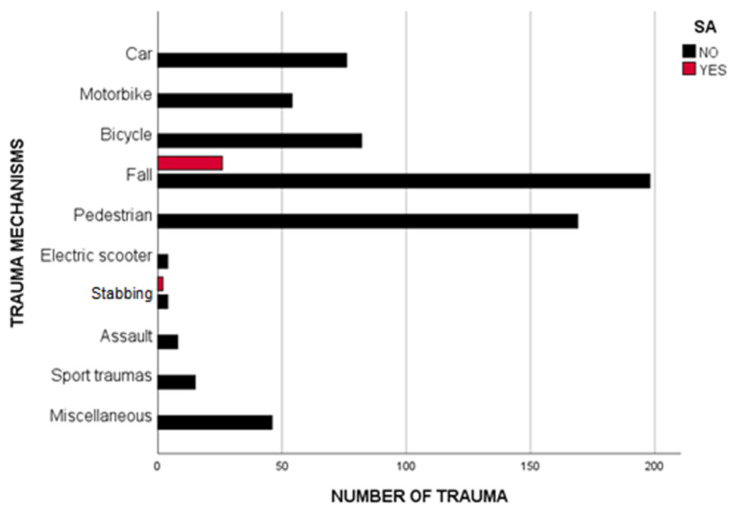
Trauma mechanisms in SAs in the overall period. Fall-related injuries were the predominant trauma mechanism of SAs in our series. The only other mechanism highlighted was self-stabbing.

**Table 1 jpm-13-00128-t001:** Baseline population characteristics.

	2015 (N = 84)	2016 (N = 98)	2017 (N = 125)	2018 (N = 119)	2019 (N = 114)	2020 (N = 144)	*p*
Sex (male), n (%)	63 (75)	70 (71.4)	83 (66.4)	77 (64.7)	70 (61.4)	90 (62.5)	0.280
Age (years)	10 (±5.4)	9.9 (±5)	9.6 (±5.2)	10 (±5.2)	9.8 (±5.1)	10.1 (±5.2)	0.977
Mechanism of trauma, n (%) - Street accident - Fall - Violence - Others	45 (53.6) 30 (35.7) 2 (2.4) 7 (8.3)	56 (57.1) 35 (35.7) 0 (0) 7 (7.1)	72 (57.6) 42 (33.6) 4 (3.2) 7 (5.6)	72 (60.5) 33 (27.8) 1 (0.8) 13 (10.9)	65 (57) 37 (32.5) 1 (0.9) 11 (9.6)	75 (52.1) 47 (32.6) 6 (4.2) 16 (11.1)	0.808 0.830 0.168 0.112
Type of trauma (blunt), n (%)	82 (97.6)	94 (95.9)	123 (98.4)	115 (96.6)	109 (95.6)	139 (96.5)	0.847
Trauma intentionality, n (%) - VRT - SA	3 (3.6) 1 (1.2)	3 (3.1) 2 (2)	8 (6.5) 5 (4)	9 (7.6) 5 (4.2)	9 (7.9) 6 (5.3)	14 (9.8) 9 (6.2)	0.020 0.029
Type of ED access, n (%) - Self-presenting	7 (8.3)	7 (7.1)	10 (8)	9 (7.6)	0 (0)	19 (13.2)	0.020
Triage code, n (%) - Green - Yellow - Red	7 (8.3) 70 (83.4) 7 (8.3)	8 (8.2) 82 (83.6) 8 (8.2)	13 (10.4) 100 (80) 12 (9.6)	9 (7.6) 95 (79.8) 15 (12.6)	5 (4.4) 82 (71.9) 27 (23.7)	36 (25) 89 (61.8) 19 (13.2)	<0.001 <0.001 0.015

**Table 2 jpm-13-00128-t002:** Clinical presentation: pre-hospital and ED vital signs.

	2015 (N = 84)	2016 (N = 98)	2017 (N = 125)	2018 (N = 119)	2019 (N = 114)	2020 (N = 144)	*p*
BP-pre (mmHg)	120 (110–130)	120 (110–130)	120 (100–130)	118 (102–130)	116 (100–127)	113 (105–130)	0.064
HR-pre (bpm)	100 (90–120)	98 (87–117)	100 (85–120)	100 (90–120)	100 (82–120)	100 (86–110)	0.620
GCS-pre	15	15	15	15	15	15	1
BP-ED (mmHg)	120 (105–130)	120 (110–125)	120 (100–130)	120 (105–127)	119 (105–127)	115 (103–130)	0.903
HR-ED (bpm)	97 (80–110)	100 (86–115)	100 (85–115)	100 (87–120)	100 (85–113)	100 (85–113)	0.775
GCS-ED	15	15	15	15	15	15	1

BP-pre: pre-hospital systolic blood pressure; HR-pre: pre-hospital heart rate; GCS-pre: pre-hospital Glasgow Coma Scale; BP-ED: emergency department systolic blood pressure; HR-ED: emergency department heart rate; GCS-ED: emergency department Glasgow Coma Scale. Results are expressed as median (interquartile range (IQR)).

**Table 3 jpm-13-00128-t003:** Anatomical distribution of major injuries (AIS15 > 2) in major trauma according to trauma mechanism.

	Head	Chest	Abdomen	Extremities
Fall (n = 54)	29	31	18	20
Car (n = 11)	5	6	1	5
Motorbike (n = 16)	3	6	3	7
Bicycle (n = 13)	6	5	3	1
Pedestrian (n = 26)	15	14	3	7
Assault (n = 1)	1	0	0	0
Stab wound (n = 1)	0	0	1	0
Sports trauma (n = 4)	3	0	1	0
Miscellaneous (n = 4)	2	0	0	2

**Table 4 jpm-13-00128-t004:** Ages 0–12 y vs. 13–18 y: trauma intentionality and mechanisms over time.

		2015		2016		2017		2018		2019		2020		*p*
		(n = 84)		(n = 98)		(n = 125)		(n = 117)		(n = 119)		(n = 144)		
		<13 y	≥13 y	<13 y	≥13 y	<13 y	≥13 y	<13 y	≥13 y	<13 y	≥13 y	<13 y	≥13 y	
		(n = 51)	(n = 33)	(n = 62)	(n = 36)	(n = 79)	(n = 46)	(n = 74)	(n = 43)	(n = 69)	(n = 45)	(n = 80)	(n = 63)	
Intentionality, n (%)	Accident	50 (98)	31 (93.9)	61 (98.4)	34 (94.4)	77 (97.5)	40 (87)	71 (95.9)	37 (86)	66 (95.7)	39 (86.7)	80 (100)	49 (77.8)	0.06
	VRT	1 (2)	2 (6.1)	1 (1.6)	2 (5.6)	2 (2.5)	6 (13)	3 (4.1)	6 (14)	3 (4.3)	6 (13.3)	0 (0)	14 (22.2)	0.02
	SA	1 (2)	0 (0)	0 (0)	2 (5.6)	1 (1.3)	4 (11.5)	0 (0)	5 (11.6)	1 (1.4)	5 (11.1)	0 (0)	9 (14.3)	0.03
Mechanisms, n (%)	Car	11 (21.6)	1 (3)	7 (11.3)	2 (5.6)	12 (15.2)	3 (6.5)	5 (6.8)	1 (2.3)	11 (15.9)	3 (6.7)	9 (11.3)	8 (12.7)	0.86
	Motorbike	1 (2)	6 (18.2)	0 (0)	6 (16.7)	3 (3.8)	7(15.2)	1 (1.4)	14 (32.6)	0 (0)	8 (17.8)	0 (0)	8 (12.7)	0.66
	Bicycle	0 (0)	1 (3)	6 (9.7)	7 (19.4)	6 (7.6)	12 (26.1)	6 (8.1)	8 (18.6)	7 (10.1)	7 (15.6)	13 (16.3)	8 (12.7)	0.03
	Electric scooter	0 (0)	0 (0)	0 (0)	0 (0)	0 (0)	0 (0)	0 (0)	0 (0)	0 (0)	0 (0)	1 (1.3)	3 (4.8)	0.01
	Pedestrian	13 (25.5)	12 (36.4)	17 (27.4)	11 (30.6)	21 (26.6)	8 (17.4)	26 (35.1)	9 (20.9)	19 (27.5)	10 (22.2)	12 (15)	11 (17.5)	0.03
	Fall	22 (43.1)	8 (24.2)	29 (46.8)	6 (16.7)	33 (41.8)	9 (19.6)	25 (33.8)	8 (18.6)	28 (40.6)	9 (20)	30 (37.5)	17 (27)	0.49
	Stabbing	0 (0)	1 (3)	0 (0)	0 (0)	0 (0)	1 (2.2)	0 (0)	0 (0)	0 (0)	0 (0)	0 (0)	4 (6.3)	0.18
	Assault	0 (0)	1 (3)	0 (0)	0 (0)	1 (1.3)	2 (4.3)	0 (0)	1 (2.3)	0 (0)	1 (2.2)	1 (1.3)	1 (1.6)	0.83
	Sport trauma	0 (0)	2 (6.1)	2 (3.2)	1 (2.8)	1 (1.3)	3 (6.5)	3 (4.1)	1 (2.3)	0 (0)	1 (2.2)	0 (0)	1 (1.6)	0.15
	Other	4 (7.8)	1 (3)	1 (1.6)	3 (8.3)	2 (2.5)	1 (2.2)	8 (10.8)	1 (2.3)	4 (5.8)	6 (13.3)	14 (17.5)	1 (1.6)	0.02

**Table 5 jpm-13-00128-t005:** Anatomical distribution and severity of trauma: 0–12 years vs. 13–18 years comparison.

		0–12 (n = 415)	13–18 (n = 269)	*p*
Head-AIS15, n (%)	1	99 (23.8)	28 (10.4)	<0.001
	2	45 (10.8)	51 (18.9)	
	3	22 (5.3)	21 (7.8)	
	4	9 (2.2)	7 (2.6)	
	5	4 (0.9)	10 (3.7)	
Chest-AIS15, n (%)	1	0 (0)	0 (0)	0.195
	2	8 (1.9)	15 (5.6)	
	3	20 (4.8)	21 (7.8)	
	4	5 (1.2)	13 (4.8)	
	5	0 (0)	3 (1.1)	
Abdomen-AIS15, n (%)	1	0 (0)	1 (0.4)	0.272
	2	15 (3.6)	25 (9.3)	
	3	8 (1.9)	10 (3.7)	
	4	3 (0.7)	6 (2.2)	
	5	0 (0)	7 (2.6)	
Extremity-AIS15. N (%)	1	8 (1.9)	16 (5.9)	0.042
	2	64 (15.4)	50 (18.6)	
	3	18 (4.3)	32 (11.9)	
	4	5 (1.2)	8 (2.9)	
	5	1 (0.2)	4 (1.5)	

## Data Availability

Data related to the study are available upon request to the corresponding author.

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
