# Peer review of "Increasing Trend in Violence-Related Trauma and Suicide Attempts among Pediatric Trauma Patients: A 6-Year Analysis of Trauma Mechanisms and the Effects of the COVID-19 Pandemic"

_jpm, 2023, doi:10.3390/jpm13010128_

Round 1

Reviewer 1 Report

Dear Maina Cecilia,

Thank you for the opportunity to review this manuscript.

Manuscript jpm-1998273 entitled “Increasing trend of violence-related trauma and suicide attempts among paediatric trauma patients: a 6 years analysis of trauma mechanisms and the effects of the Covid-19 pandemic” has been reviewed. This manuscript assessed the annual trends in the different mechanisms of injury and the effects of the coronavirus disease-2019 (COVID-19) pandemic among pediatric trauma patients. While the findings presented in this study seem to be of interest to this journal’s audience, I have various concerns that need to be addressed prior to the publication of this manuscript. I apologize for being critical in my review, but hopefully, my comments will aid in improving your manuscript and getting it published.

1.             It was stated that this study aimed to analyze the trend over time of pediatric trauma patients, focusing on the change in trauma mechanisms and intentionality, and the effect of the COVID-19 pandemic (Lines 48-51). However, the results of the comparison analysis between the pre-pandemic group admitted from 2015 to 2019 versus the pandemic group admitted in 2020 were not clearly shown in your manuscript.

2.             The Methods section does not sufficiently explain the methodology and statistical analyses used in your study. In particular, it is unclear why the authors focused on the patient’s age year as a subclass analysis. Moreover, it is necessary to explain why the pediatric patients were divided into two groups, those younger and older than 13 years old.

Defining the key terms (ex: major trauma) used in your manuscript is necessary. In the Materials and Methods section, patients with an AIS score >2 were classified as those who experienced major trauma. Major trauma patients have an ISS score ≥9 because ISS is calculated by squaring the AIS score. Kindly explain why you defined major trauma patients as those with an ISS score of ≥12.

Author Response

Thanks for reviewing so carefully our manuscript.

We will reply per each point to your suggestions and remarks.

- It was stated that this study aimed to analyze the trend over time of pediatric trauma patients, focusing on the change in trauma mechanisms and intentionality, and the effect of the COVID-19 pandemic (Lines 48-51). However, the results of the comparison analysis between the pre-pandemic group admitted from 2015 to 2019 versus the pandemic group admitted in 2020 were not clearly shown in your manuscript.

Thanks for your comment. We agree with your point and modified with a more clear statement in the results. Line 130 to 134.

- The Methods section does not sufficiently explain the methodology and statistical analyses used in your study. In particular, it is unclear why the authors focused on the patient’s age year as a subclass analysis. Moreover, it is necessary to explain why the pediatric patients were divided into two groups, those younger and older than 13 years old.

Thanks again for your consideration.

We modified the methods section to make it more clear and straightforward.

We decided to explore the differences between patients younger or older than 13 years since recent reports showed how suicide attempts were more frequent in patients older than 13 years old with a median of 16 years. Our clinical experience was also similar to the published report. Therefore, as reported, we hypothesized that also in our cohort the mechanisms and injury patterns were different in patients older than 13 years old.

Reference number 23 (Theodorou CM, Yamashiro KJ, Stokes SC, Salcedo ES, Hirose S, Beres AL. Pediatric suicide by violent means: a cry for help and a call for action. Inj Epidemiol [Internet]. 2022;9(1):1–5. Available from: https://doi.org/10.1186/s40621-022-00378-6)

- Defining the key terms (ex: major trauma) used in your manuscript is necessary. In the Materials and Methods section, patients with an AIS score >2 were classified as those who experienced major trauma. Major trauma patients have an ISS score ≥9 because ISS is calculated by squaring the AIS score. Kindly explain why you defined major trauma patients as those with an ISS score of ≥12.

Thanks a lot for your point. The organ-specific Abbreviated Injurt Scale 2015 Revision (2015) was used to describe the anatomical distribution of injuries in trauma patients. An AIS15 score > 2 recognized a severely injured anatomical district. The comprehensive trauma severity was expressed by Injury Severity Score (ISS), obtained from the sum of the highest three squared AIS15 score. Major trauma were defined by an ISS ≥ 12.

We adopted this definition following the more recent evidences as summarized in reference number 12. (Van Ditshuizen JC, Sewalt CA, Palmer CS, Van Lieshout EMM, Verhofstad MHJ, Den Hartog D, et al. The definition of major trauma using different revisions of the abbreviated injury scale. Scand J Trauma Resusc Emerg Med. 2021;29(1):1–10)

Reviewer 2 Report

The study presents interesting observations around injury frequency and patterns - interpreted in the context of COVID 19

whilst this can easily be used to explain a sudden increase in non accidental trauma and suicide, I can not find any explanation why there is a similar sudden increase from 2015 - 2016. I think, readers would have the same question and would appreciate an explanation for this increase as well.

Author Response

The study presents interesting observations around injury frequency and patterns - interpreted in the context of COVID 19 whilst this can easily be used to explain a sudden increase in non-accidental trauma and suicide, I cannot find any explanation why there is a similar sudden increase from 2015 - 2016. I think, readers would have the same question and would appreciate an explanation for this increase as well.

Thanks a lot for your point. The slight increase reported from 2015/2016 is referred to the increasing experience of our center in pediatric trauma management since during those years we have been identified as level one trauma center with pediatric commitment. We added at line 222 to 224 a brief explanation on that.

Round 2

Reviewer 1 Report

Thank you for the opportunity to review this manuscript.

This second version of the paper has been much improved as per my comments, which I believe the paper will be of interest of the readership of Journal of Personalized Medicine.